# Machine learning model based on radiomics features for AO/OTA classification of pelvic fractures on pelvic radiographs

Jun Young Park[1], Seung Hwan Lee[2,3], Young Jae Kim[1,4,5], Kwang Gi Kim[1,4,5]*, Gil Jae Lee[2,3]*

1 Department of Health Sciences and Technology, Gachon Advanced Institute for Health Sciences and Technology (GAIHST), Gachon University, Incheon, Republic of Korea, 2 Department of Trauma Surgery, Gachon University Gil Medical Center, Gachon University, Incheon, Republic of Korea, 3 Department of Traumatology, Gachon University College of Medicine, Gachon University, Incheon, Republic of Korea, 4 Department of Medical Devices R&D Center, Gachon University Gil Medical Center, Gachon University, Incheon, Republic of Korea, 5 Department of Biomedical Engineering, Pre-medical Course, College of Medicine, Gachon University, Incheon, Republic of Korea

☯ These authors contributed equally to this work.
* kimkg@gachon.ac.kr (KGK); nonajugi@gilhospital.com (GJL)

**Data Availability Statement:** The training data, specifically the image data, cannot be shared publicly due to the nature of medical data and as this study was conducted with the data that include

## Abstract

Depending on the degree of fracture, pelvic fracture can be accompanied by vascular damage, and in severe cases, it may progress to hemorrhagic shock. Pelvic radiography can quickly diagnose pelvic fractures, and the Association for Osteosynthesis Foundation and Orthopedic Trauma Association (AO/OTA) classification system is useful for evaluating pelvic fracture instability. This study aimed to develop a radiomics-based machine-learning algorithm to quickly diagnose fractures on pelvic X-ray and classify their instability. data used were pelvic anteroposterior radiographs of 990 adults over 18 years of age diagnosed with pelvic fractures, and 200 normal subjects. A total of 93 features were extracted based on radiomics:18 first-order, 24 GLCM, 16 GLRLM, 16 GLSZM, 5 NGTDM, and 14 GLDM features. To improve the performance of machine learning, the feature selection methods RFE, SFS, LASSO, and Ridge were used, and the machine learning models used LR, SVM, RF, XGB, MLP, KNN, and LGBM. Performance measurement was evaluated by area under the curve (AUC) by analyzing the receiver operating characteristic curve. The machine learning model was trained based on the selected features using four feature-selection methods. When the RFE feature selection method was used, the average AUC was higher than that of the other methods. Among them, the combination with the machine learning model SVM showed the best performance, with an average AUC of 0.75±0.06. By obtaining a feature-importance graph for the combination of RFE and SVM, it is possible to identify features with high importance. The AO/OTA classification of normal pelvic rings and pelvic fractures on pelvic AP radiographs using a radiomics-based machine learning model showed the highest AUC when using the SVM classification combination. Further research on the radiomic features of each part of the pelvic bone constituting the pelvic ring is needed.

sensitive personal information.Therefore, we are unable to open the dataset that was used for training with the imposing of the Institutional Review Board. Data used for training are available from the ethics committee (contact via email: irb@gilhospital.com) for researchers who meet the criteria for access to confidential data.

**Funding:** This work was supported by the Korea Medical Device Development Fund grant funded by the Korea government (the Ministry of Science and ICT, the Ministry of Trade, Industry and Energy, the Ministry of Health & Welfare, the Ministry of Food and Drug Safety) (Project Number: 1711196789, RS-2023-00252804). The funders had no role in study design, data collection and analysis, decision to publish, or preparation of the manuscript.

**Competing interests:** The authors have declared that no competing interests exist.

## Introduction

Many bones form joints in the pelvic ring, and many blood vessels are distributed. The ring also protects the internal organs in the pelvic cavity [1]. Fracture of the pelvic ring can lead to hemorrhagic shock, accompanied by damage to nearby organs and severe bleeding [2]. Pelvic fractures require prompt diagnosis and treatment [3]. In the early stages of trauma, pelvic radiographs help obtain information such as the degree and location of the fracture relatively quickly. However, the diagnosis of fractures can be difficult because the quality of radiographic images can be inadequate depending on the patient's condition or imaging environment [4]. However, computed tomography (CT) scans in hemodynamically unstable patients may worsen their condition.

Increasing the diagnostic utilization of radiography, which can be performed relatively easily and quickly in the early stages of trauma, may be helpful for emergency patients. Recently, studies using artificial intelligence have been reported on patients with various emergency diseases who visit the emergency room [5, 6]. A deep learning model was developed to predict the exacerbation of COVID-19 pneumonia and diagnose pneumothorax, pleural effusion, and fracture in chest X-ray images, and the developed model showed a similar or higher area under the curve (AUC) than the diagnostic accuracy of specialists [5, 6]. In addition to chest X-ray images, studies to detect femoral neck fractures using other X-ray images and to classify displaced and non-displaced fractures have also been reported [7, 8]. Another study proposed an automatic fracture detection system based on the Ada-ResNet backbone network in various X-ray images [9].

One study reported the diagnosis of osteoporosis on lumbar CT images using a radiomics-based machine learning model rather than a deep learning model [10]. A study to predict osteoporosis in the femoral region using three-dimensional radiomic features in abdominopelvic CT images has also been reported [11]. Most studies using radiomic features were aimed at predicting osteoporosis, reduction in bone strength, and structural weakening, as in previous research, and no studies have classified patterns or types of fractures.

This study attempted to classify pelvic fracture patterns according to the Association for Osteosynthesis Foundation and Orthopedic Trauma Association (AO/OTA) classification system for pelvic X-ray images using a radiomics-based machine-learning algorithm [12]. The AO/OTA classification system is useful for evaluating pelvic instability in patients with pelvic fractures. A radiomics-based machine-learning algorithm was developed to rapidly diagnose fractures in pelvic X-ray images and classify pelvic instability.

## Materials and methods

### Data

Between January 2015 and December 2020, 990 adults aged 18 years or older who visited the regional trauma center of Gachon University Gil Hospital and were diagnosed with pelvic fracture and 200 patients without pelvic fracture each received a pelvic radiograph before and after imaging (an anteroposterior pelvic radiograph [hereafter referred to as pelvic AP X-ray] was collected and used for data analysis). The study was conducted in accordance with the Declaration of Helsinki and approved by the Institutional Review Board of Gachon University Gil Medical Center, which waived the equirement for informed consent from participants (IRB NO. GAIRB2022-153). The data access date for research purposes began on January 15, 2023, and continued until the end of the study. For the 990 cases of pelvic fracture diagnosed as pelvic fracture, (1) the radiologist's interpretation of the pelvic AP X-ray, (2) the orthopedic specialist's opinion, and (3) the pelvic area CT image and interpretation findings were evaluated

**Table 1. Classification of pelvic ring fractures according to AO/OTA classification system.**

| Type A | Rotationally and vertically stable, the sacroiliac complex is intact. Type A fractures are mostly managed non-operatively |
|---|---|
| Type B | Rotationally unstable and vertically stable, caused by external or internal rotational forces, results in partial disruption of the posterior sacroiliac complex |
| Type C | Rotationally unstable and vertically unstable, complete disruption of the posterior sacroiliac complex. These unstable fractures are mostly caused by high-energy trauma like falls from height, motor vehicle accidents or crushing injuries |

by a trauma surgeons with more than 10 years of experience, who confirmed all fracture sites on the pelvic AP X-ray. In addition, the pelvic AP X-ray images of 990 patients diagnosed with pelvic fractures included A-type (A1, A2, A3), B-type (B1, B2, B3) and C-type (C1, C2, C3). Table 1 is a description of pelvic fractures Type A, Type B, and Type C classified according to the AO/OTA classification system.

It was classified into 1,190 images: 564 images of type A, 252 images of type, 174 images of type C and 200 normal images (Table 2). The procedure was performed by a single trauma surgeons, based on pelvic AP X-ray imaging findings, pelvic CT findings, and medical records previously classified by orthopedic surgeons according to the AO/OTA system.

In this study, the features were extracted only from the pelvic ring region. Fig 1 shows the region of interest (ROI) for the pelvic region. The ROI for the pelvic region was defined by a trauma surgeon with more than 10 years of experience checking AP X-ray images of the pelvis. The defined pelvic region ROI included the left and right Ililum, Pubic Bone, and Ischium. The software used to obtain ROI was AVIEW (Corelinesoft, Seoul, Republic of Korea). A mask image in the form of Fig 1 (b) was obtained according to the pelvic region defined using AVIEW.

From the 990 pieces of collected data, images with severe blurring or covering the pelvic area due to organs or gases were excluded. Fig 2 shows examples of the excluded images. The A-, B-, and C-type pelvic fracture images and normal pelvic image data were divided into training and test image datasets at a ratio of 7:3 (Table 3). Type A training and testing data are presented in Chapters 387 and 166, Type B training and testing data in Chapters 173 and 74, Type C learning and testing data in Chapters 120 and 52, and the normal training and test data in Chapters 138 and 59. Different weights were assigned to each class for learning to overcome the data imbalance problem.

**Table 2. The number of X-ray data and normal data collected by pelvic fracture type.**

| Type | 2015 | 2016 | 2017 | 2018 | 2019 | 2020 | Total |
|---|---|---|---|---|---|---|---|
| A1 | 3 | 2 | 5 | 5 | 2 | 1 | 18 |
| A2 | 69 | 61 | 83 | 108 | 91 | 115 | 527 |
| A3 | 3 | 1 | 4 | 2 | 4 | 5 | 19 |
| B1 | 1 | 14 | 21 | 22 | 25 | 17 | 100 |
| B2 | 31 | 18 | 9 | 16 | 16 | 33 | 123 |
| B3 | 4 | 5 | 5 | 6 | 6 | 3 | 29 |
| C1 | 13 | 13 | 21 | 11 | 17 | 16 | 91 |
| C2 | 6 | 5 | 9 | 11 | 7 | 3 | 41 |
| C3 | 5 | 5 | 4 | 11 | 10 | 7 | 42 |
| Normal | 30 | 27 | 28 | 31 | 32 | 52 | 200 |
| Total | 135 | 124 | 161 | 192 | 178 | 400 | 1,190 |

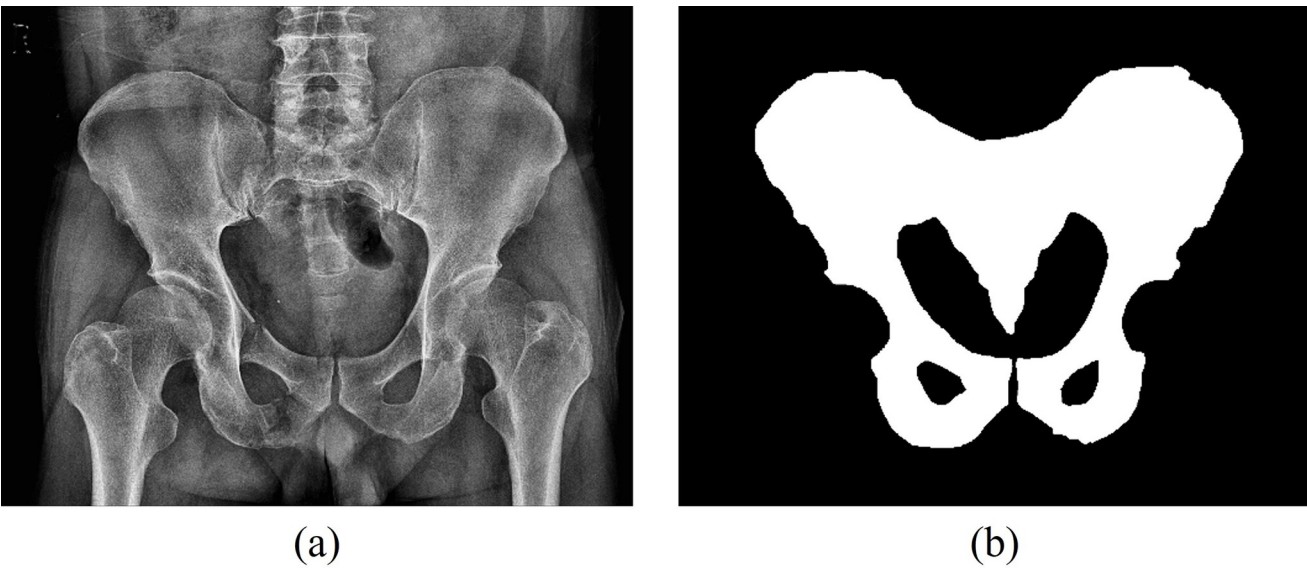

(a)                                                                                       (b)

**Fig 1.** (a) a collection pelvic X-ray image, (b) a pelvic region ROI image.

### Research environment

The experiment in this study used a system consisting of an NVIDIA GeForce GTX 1050 Ti (NVIDIA, Santa Clara, CA, USA) graphics processing unit, an Intel® core™ i7-10700 (Intel, Santa Clara, CA, USA) CPU, and 16GB RAM, and was performed using the Windows 10 Pro operating system. The libraries used in the experiment were a Python open-source library (PyRadiomics 3.0.1) [13], a library focused on real-time image processing (OpenCV-Python

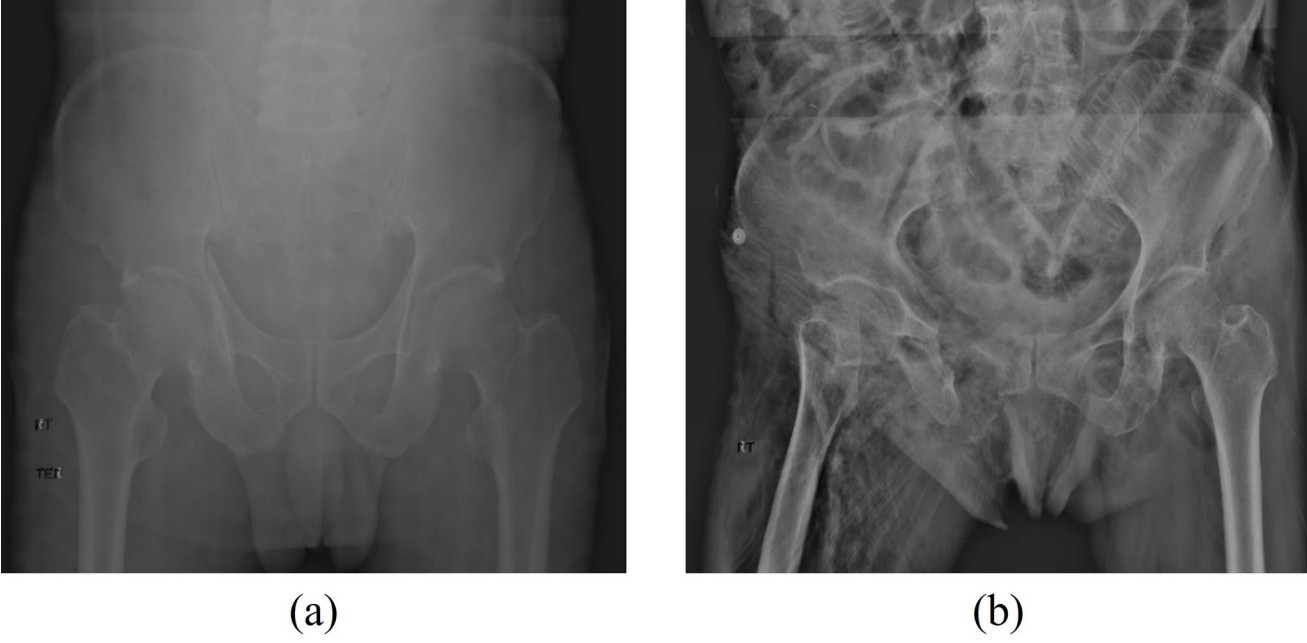

(a)                                                                                       (b)

**Fig 2.** (a) in case the pelvic area is not clearly visible due to severe blur and (b) in case the pelvic area is not clearly visible due to organs and gases.

**Table 3. Datasets used for training and testing of machine learning models.**

|  | Type A | Type B | Type C | Normal | Total |
|---|---|---|---|---|---|
| **Train set** | 387 | 173 | 120 | 138 | 818 |
| **Test set** | 166 | 74 | 52 | 59 | 351 |
| **Total** | 553 | 247 | 172 | 197 | 1,169 |

4.6.0.66), a Python-based open-source machine learning package (scikit-learn 0.24.2), and a Python library (Matplotlib 3.3.4) that can plot data in various forms.

## Data preprocessing

Among the collected pelvic AP X-ray image data, there were images that were too dark or bright depending on the shooting environment, making it difficult to sufficiently identify the fracture site. Because these image data can be unfavorable for effective learning by machine learning algorithms, a histogram equalization process was performed before the analysis.

Histogram equalization is an image-processing method that equalizes the brightness values of an image to improve its contrast. It helps obtain a clearer image by restoring the contrast lost in the image and retaining the brightness value [14]. Fig 3 shows that the pelvic AP X-ray image with low contrast was converted to a high contrast image through the histogram equalization process. The horizontal axis of the histogram represents the intensity value of the image, and the vertical axis indicates the frequency at which the corresponding intensity value is used. If the histogram of the image is distributed in a narrow area, the contrast is low, and the image is not clear, whereas if the histogram is spread uniformly over a wide area, the contrast is high, and the image is clear.

## Feature extraction

The features for each region were extracted through radiomics, which can express quantitative values by extracting mathematical and statistical information from imaging characteristics that are difficult to check visually in medical images (Fig 4). Eighteen first-order features were obtained by analyzing the histogram of the image, and 75 second-order features were obtained by calculating the correlation between adjacent pixels of the image. Second-order features include 24 Gray Level Co-occurrence Matrices (GLCM) that identify the relationship between pixels, 16 Gray Level Run Length Matrices (GLRLM) that find pixels with the same gray level value, and the size and 16 Gray Level Size Zone Matrix (GLSZM) to analyze intensity, 5 Neighboring Gray Tone Difference Matrix (NGTDM) to correlate gray level values between pixels, and Gray Level Dependence Matrix (to determine gray level dependencies in images) (GLDM). A total of 93 features were extracted using the two feature extraction methods mentioned above [15–18].

## Feature selection method and machine learning model

In this study, significant features were selected using a feature-selection method that targeted features extracted through radiomics. Feature selection can improve the learning performance and accuracy of machine learning models, and reduce computational complexity and overfitting [19]. There are three feature-selection methods: the filter, wrapper, and embedded methods. The filter method determines the correlation between features and then selects those with a high correlation coefficient. The filter methods included Pearson's correlation, analysis of variance (ANOVA), and chi-squared test [20]. The wrapper method extracts the feature subset that exhibits the best performance in terms of prediction accuracy. Although the time and cost

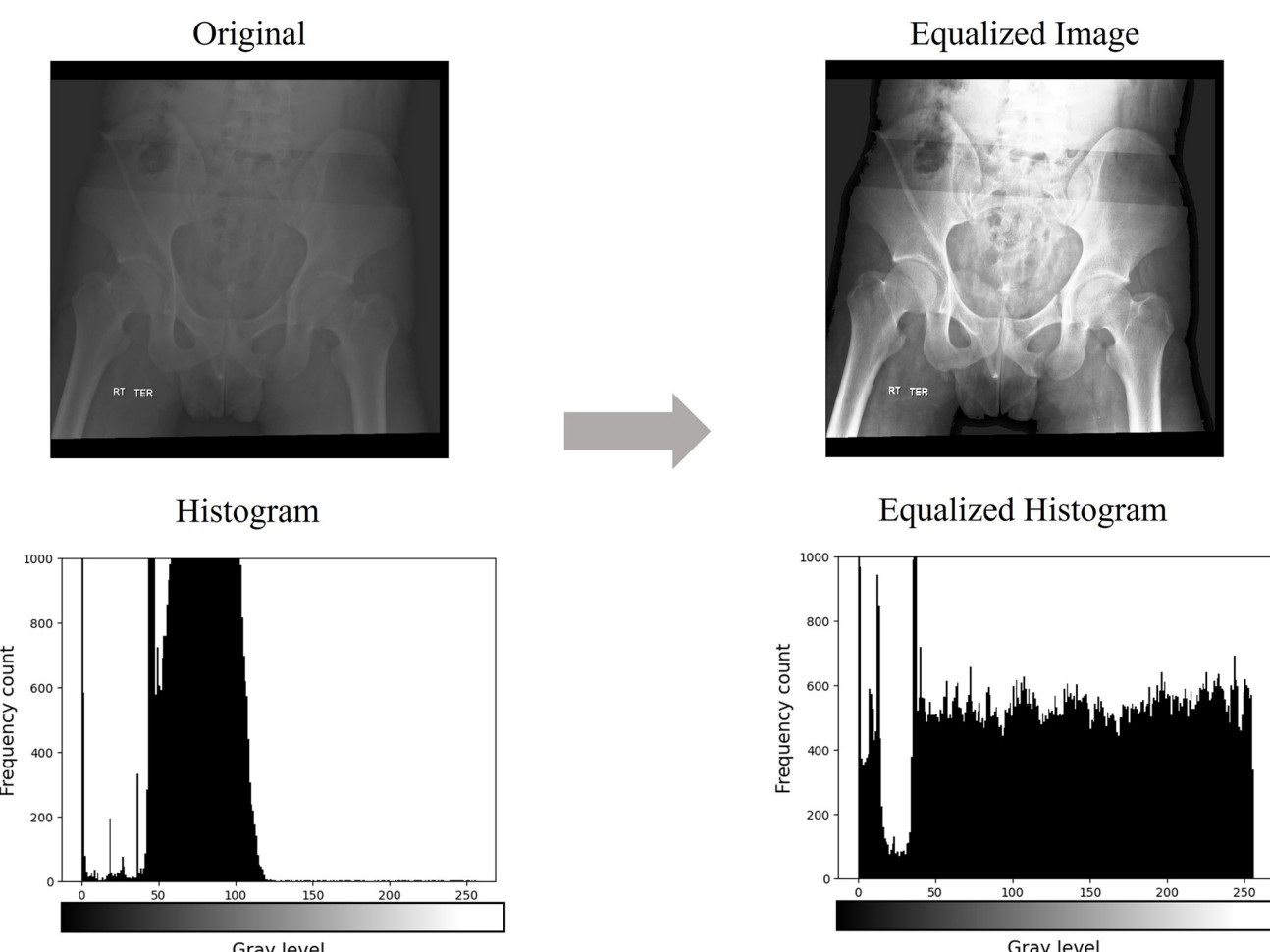

**Fig 3. Image and histogram changes before and after application of the histogram equalization algorithm.**

are relatively high, it can be the most suitable method for model performance. Wrapper methods include forward selection, backward elimination, and RFE [21]. The embedded method removes unnecessary features by gradually selecting them while learning the model several times. Embedded methods include least absolute shrinkage and selection operator (LASSO), which is constrained through the L1-norm, and the Ridge method, which is constrained by the L2-norm [22].

In this study, the RFE, SFS, LASSO, and Ridge feature selection methods were used. Logistic regression assumes a linear relationship between independent and dependent variables and estimates regression coefficients from data with the goal of accurately classifying fracture types (Type A, Type B, and Type C) and normal areas in pelvic AP X-ray image regression; a support vector machine that finds the hyperplane with the largest margin to classify each data class; a random forest that forms multiple decision trees with an ensemble machine learning model and randomly selects features; and a tree-based transformation of the gradient boosting algorithm. Ensemble machine learning model extreme gradient boosting, a multi-layer perceptron that is a form of artificial neural network, includes at least one nonlinear hidden layer, k, that determines the class to which the data belongs by looking at the nearest k surrounding data through Euclidean distance calculation. A total of 7 machine learning models of light gradient

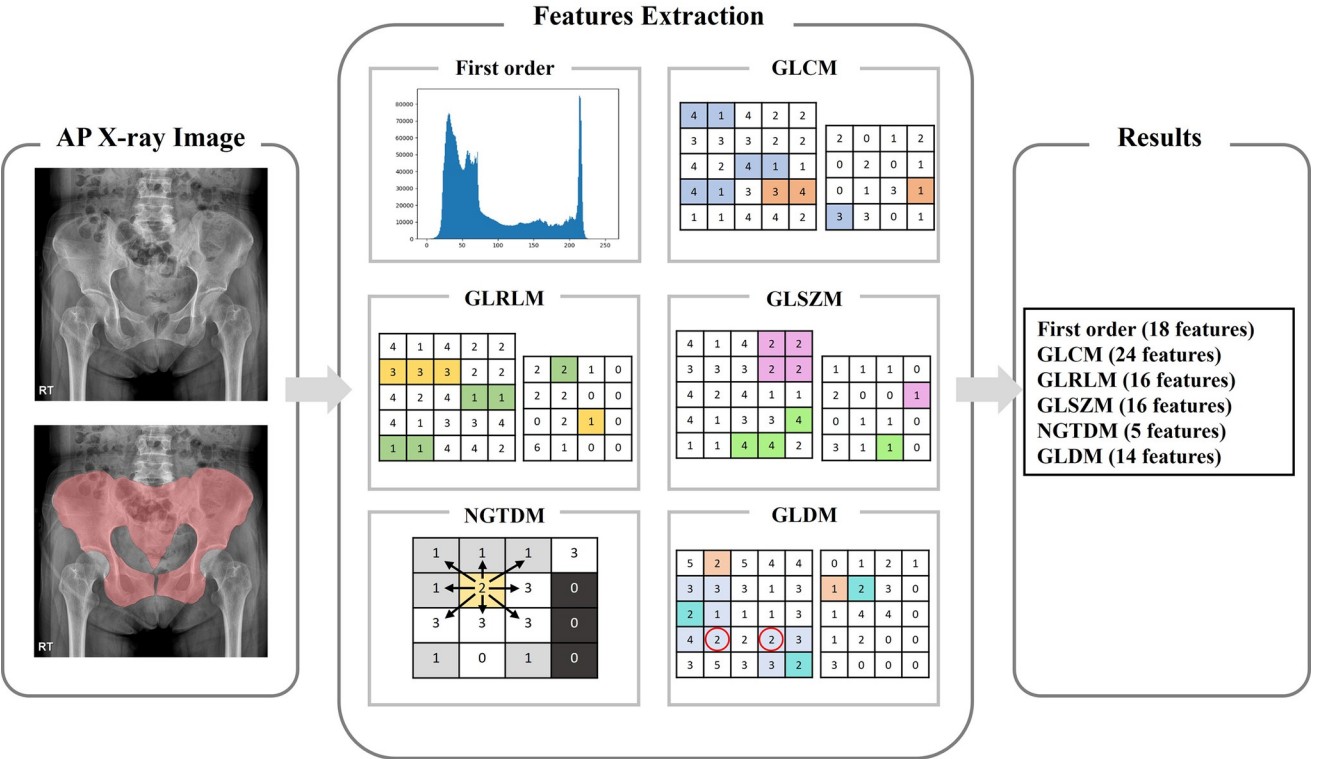

**Fig 4. Radiomics feature extraction is performed for the pelvic ring region.** 18 first-order features, 75 second-order features. Second-order features included 24 GLCM, 16 GLRLM, 16 GLSZM, 5 NGTDM, and 14 GLDM.

boosting machine, which is a modified model of the gradient boosting algorithm, were used while using leaf-based splitting instead of nearest neighbor and tree-based splitting [23–29]. In addition, to solve the multi-class classification problem, One vs Rest was used, which learns by treating one class as 1 and the remaining classes as 0.

## Statistical analysis method

In this study, the visual analysis results of the medical staff and machine learning prediction results were compared to evaluate the classification performance of pelvic fractures. The AUC was obtained using true positive (TP), false negative (FN), true negative (TN), and false positive (FP) values obtained through comparison. As shown in Fig 5, the receiver operating characteristic (ROC) curve was analyzed and the area under the ROC curve was calculated. The ROC curve shows how the true-positive rate (TPR) changes with the false-positive rate (FPR) changes. The closer the TPR is to 1, the higher the classification performance of pelvic fracture instability; the closer the FPR is to 0, the higher the classification performance.

## Results and discussion

In this study, 28 classification results were derived using four feature selection methods and seven machine-learning models. The 28 learning outcomes were compared with the classification results of the trauma surgeons. Fig 6 presents a graph expressing the AUC of each feature selection and classifier combination as a heatmap.

In Fig 7, the macro-average ROC curves are presented to analyze the average performance of the machine learning models LR, SVM, RF, XGB, MLP, KNN, and LGBM for the

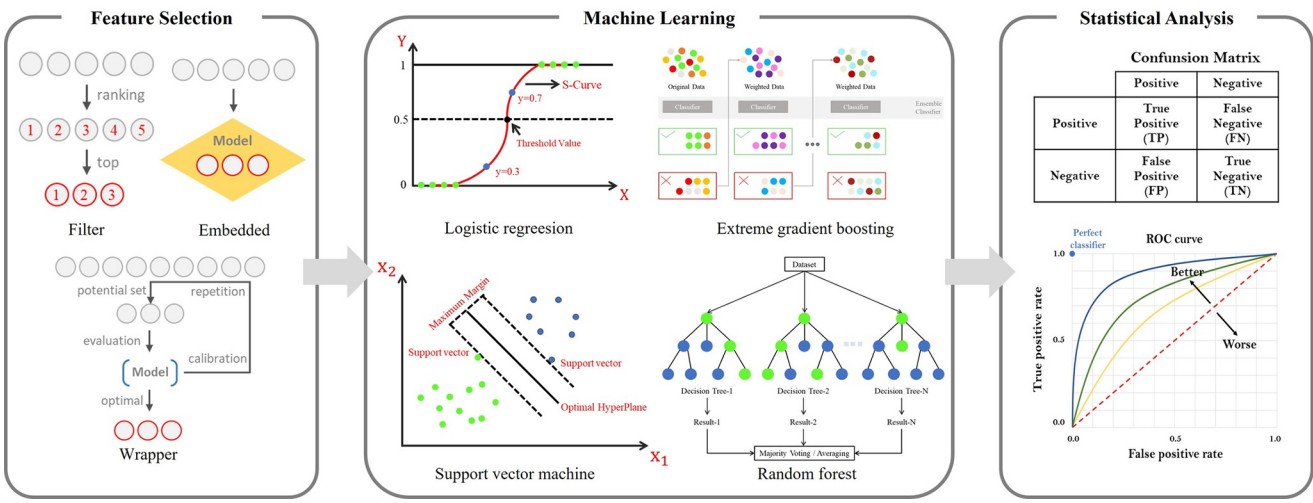

**Fig 5. After radiomics feature extraction, the feature selection method is used to select meaningful features for training the machine learning model.** Learning proceeds based on selected features. Comparison of specialist's reading findings and machine learning model prediction results.

classification of normal and pelvic fracture types. As a result of quaternary-classification, the AUCs of the micro-average ROC obtained from the four different ROC curves were 0.73, 0.75, 0.74, 0.74, 0.72, 0.67, and 0.73 for LR, SVM, RF, XGB, MLP, KNN, and LGBM, respectively.

Fig 8 shows the feature importance average values of RFE and each machine learning model combination, which had a high average performance compared to the other feature selection methods and AUC. The features with high importance that significantly affect normal and pelvic fracture type classification performance are Maximum, MaximumProbability, GrayLevelNonUniformityNormalized (GLNN), Minimum, Difference Variance, 10Percentile, Range, LowGrayLevelRunEmphasis (LGLRE), GrayLevelVariance (GLV), and 90Percentile. The features are a measure of the maximum gray-level intensity, a measure of the variability of the gray-level intensity value, a measure of the minimum gray-level intensity, a measure of pairs of different gray-level intensities that deviate more from the mean, a 10-percentile value of the gray-level intensity, and an ROI value. The gray value range, which measures the distribution of low grayscale values, measures the dispersion of gray level intensities and is the 90th-percentile value of the gray level intensities. The top three features among the ten selected features were Maximum, MaximumProbability, and GLNN.

Fig 9 shows the feature importance value for each combination of the RFE feature selection method and machine learning model. The top three significant features of the machine learning model LR were Maximum, GLNN, and MaximumProbability, with importance values of 0.104, 0.058, and 0.053, respectively. The top three significant features of the SVM machine learning model were MaximumProbability, Maximum, and Minimum, with importance values of 0.063, 0.062, and 0.035, respectively. The top three significant features of the RF machine learning model RF were Maximum, DifferenceVariance, and GLNN, with importance values of 0.051, 0.046, and 0.025, respectively. The top three significant features of the machine learning XGB model were Maximum, Minimum, and MaximumProbability, with importance values of 0.034, 0.032, and 0.030, respectively. The top three significant features of the MLP machine learning model MLP were Maximum, DifferenceVariance, and GLNN, with importance values of 0.081, 0.031, and 0.031, respectively. The top three significant features of the machine learning model KNN were Maximum, MaximumProbability, and 10Percentile, with importance values of 0.064, 0.034, and 0.030, respectively. The top three significant features of

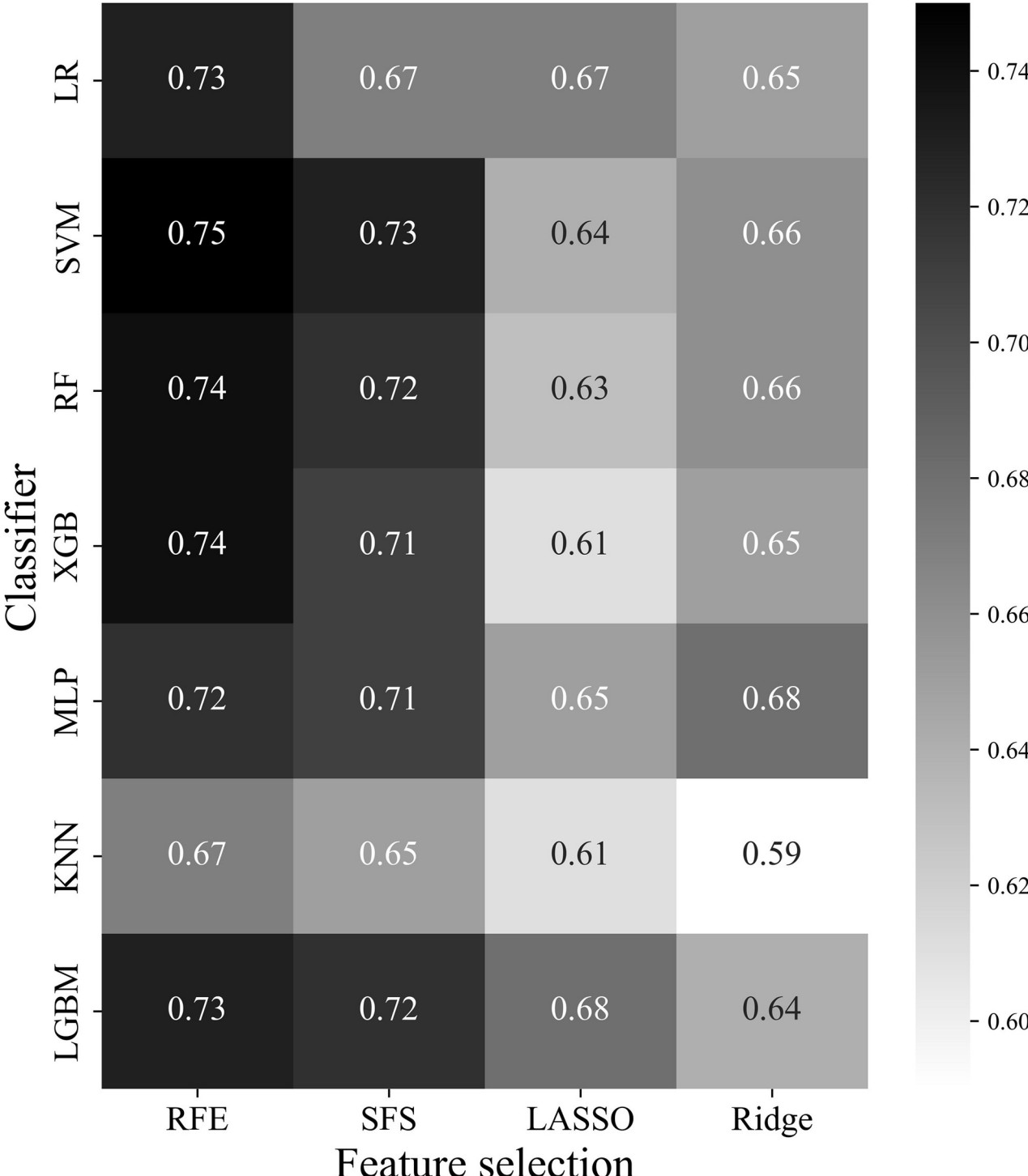

**Fig 6. AUC for each combination of LR, SVM, RF, XGB, MLP, KNN, and LGBM machine learning models and RFE, SFS, LASSO, and Ridge feature selection methods through heatmap.** The higher the heatmap value, the closer it is to black, and the higher the performance for the classification of instability of pelvic fracture. LR, logistic regression; SVM, support vector machine; RF, random forest; XGB, extreme gradient boosting; MLP, multi-layer perceptron; KNN, k-nearest neighbor; LGBM, light gradient boosting machine; RFE, recursive feature elimination; SFS, sequential feature selection.

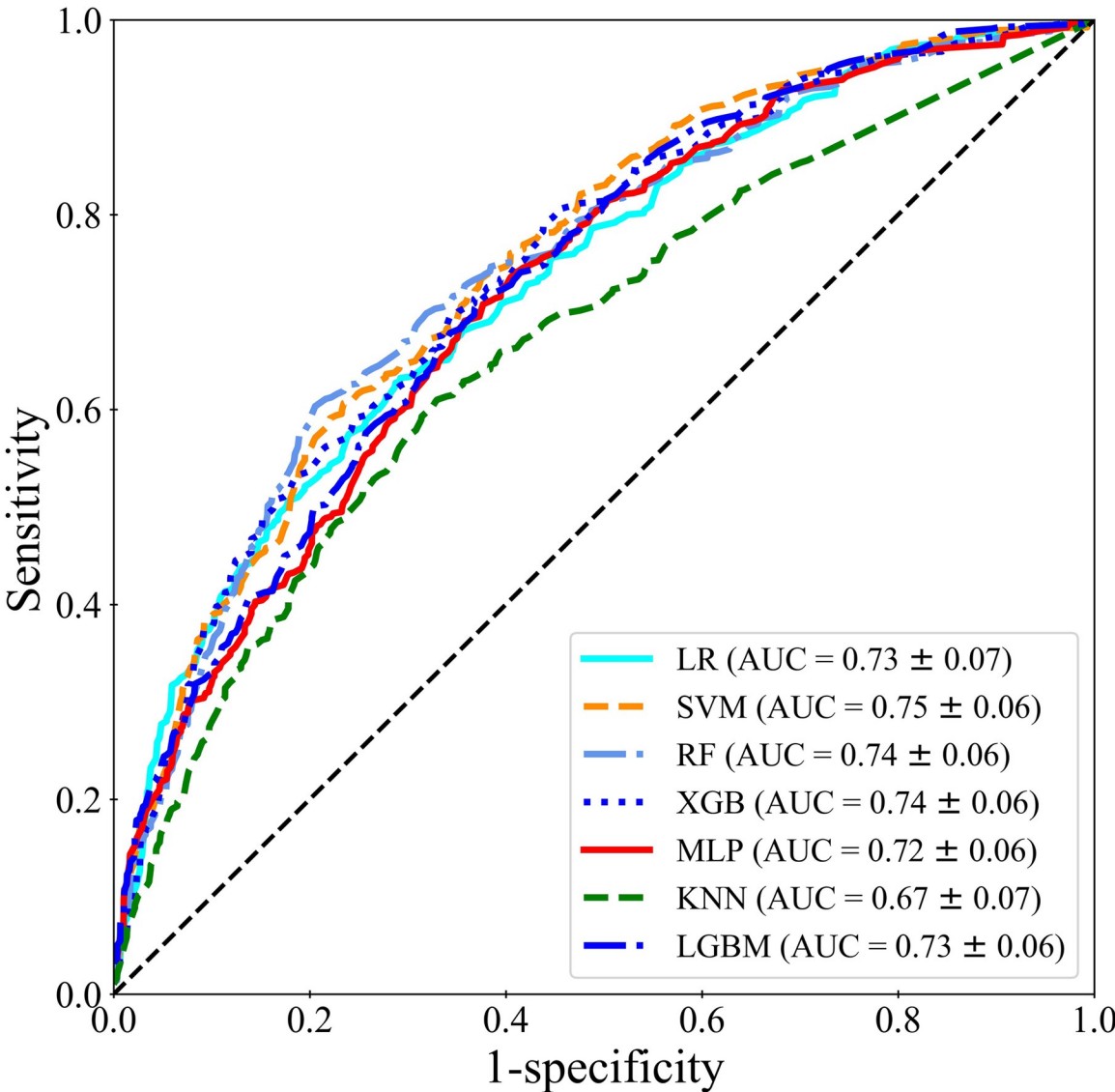

**Fig 7. ROC analysis results of machine learning models LR, SVM, RF, XGB, MLP, KNN, and LGBM for classification of normal and pelvic fracture types when using feature selection method RFE.**

the machine learning model, LGBM, were are Maximum, DifferenceVariance, and GLNN, with importance values of 0.094, 0.037, and 0.025, respectively.

As pelvic fractures can be accompanied by severe bleeding, it is important to promptly diagnose them. Depending on the fracture type, severe bleeding can be predicted during the early stages of injury. Recently, AO/OTA classification has been used to classify the fracture patterns of pelvic fractures, and it is also possible to predict whether bleeding occurs at an early stage. However, because AO/OTA classification has rather complicated classification standards, its utilization is low in emergency situations. Therefore, this study attempted to classify the AO/OTA of pelvic fractures using a machine learning model based on radiomics of pelvic X-rays.

AO/OTA classification is classified into types A, B, and C based on the location of the fracture and vertical and rotational instability of the pelvic ring. The bones that compose the pelvic ring have different structures and shapes depending on their location. For example, because

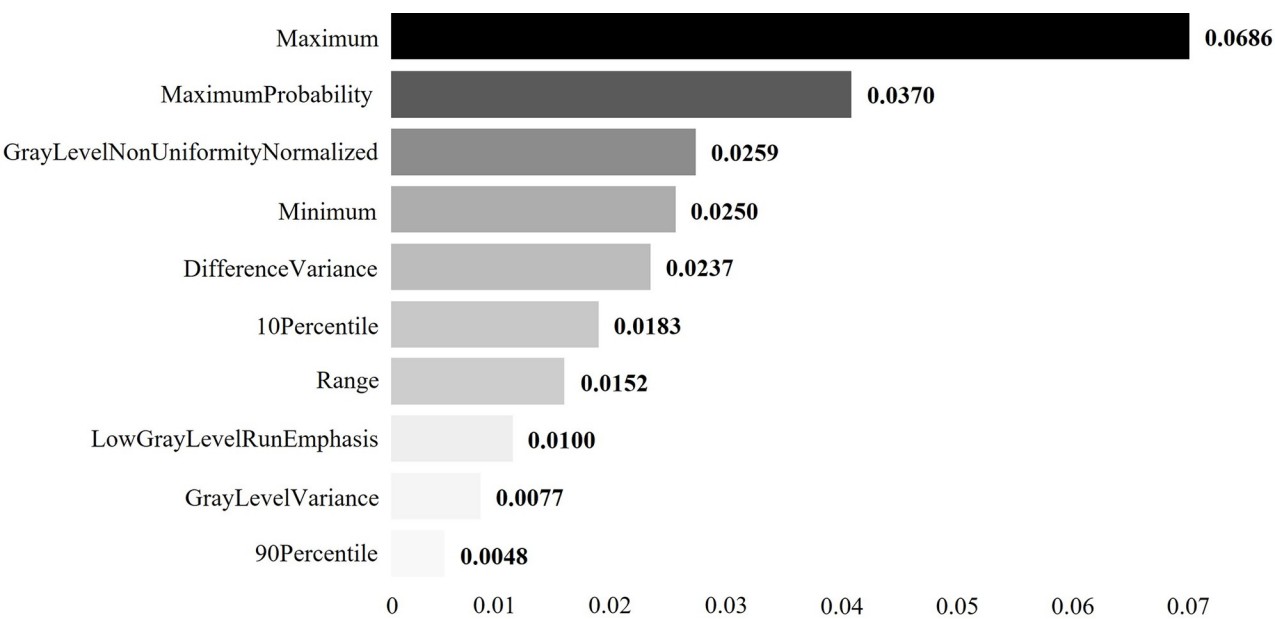

**Fig 8. A graph of the average feature importance of feature selection method RFE and each machine learning model combination.** The higher the heatmap value, the closer it is to black, and the higher the impact on the instability classification of pelvic fracture.

the pubis is rod-shaped, the ilium is plate-shaped, and the pubis and sacroiliac joints form a joint, widening of the joint may represent a feature different from a fracture. Therefore, this study proceeded with the hypothesis that the imaging features would show different features according to the fracture of each bone constituting the pelvic ring, that is, the texture of the fracture site of each bone. AO/OTA classification is made possible because the features of

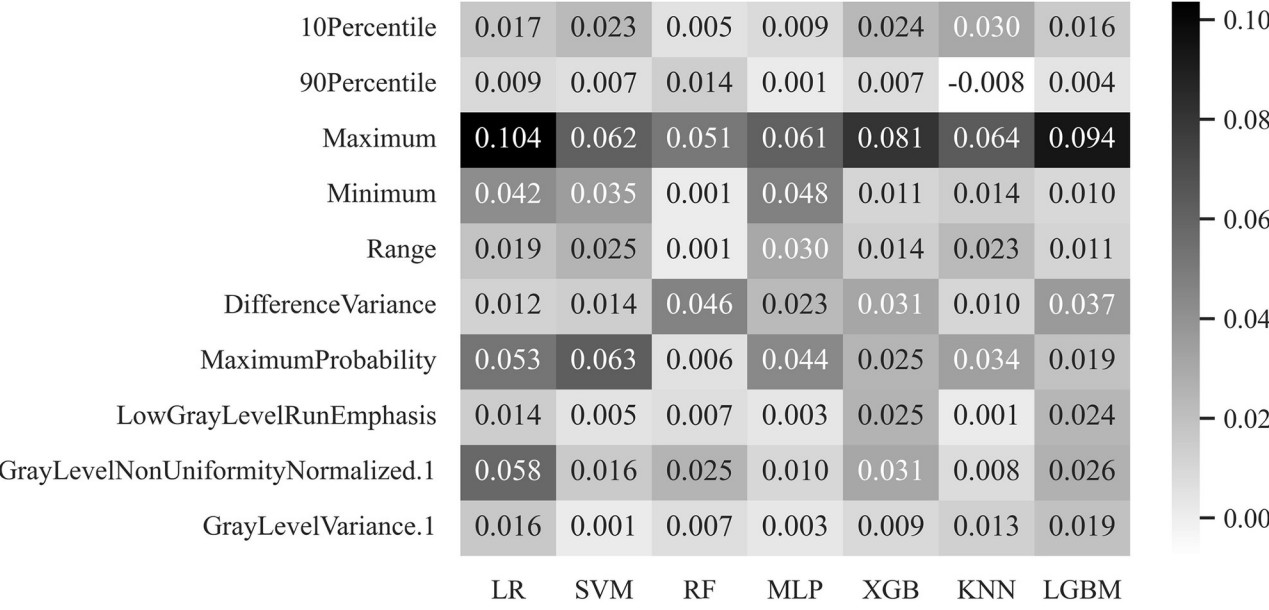

**Fig 9. Feature importance graph for feature selection method RFE and each machine learning model: LR, SVM, RF, MLP, XGB, KNN, and LGBM.** The higher the heatmap value, the closer it is to black, and the higher the impact on the instability classification of pelvic fracture.

different textures of the corresponding part, according to the fracture of each bone constituting the pelvic ring, eventually contain positional information.

In this study, radiomics-based features were extracted from X-ray images for the objective and quantitative classification of normal and pelvic fracture types, and machine learning result analysis and comparison were performed using four feature selections and seven classifiers. To evaluate the feasibility of using the features selected using feature selection, the classification performance of normal and pelvic fracture types was evaluated by comparing and analyzing each of the seven machine learning prediction results and the medical staff's reading results. RF showed the best classification performance, with an AUC of 0.73. In the SVM classifier with the highest AUC, MaximumProbability, Maximum, and Minimum were the top three features that were significant for pelvic normality and fracture type classification. Among 28 combinations of feature selection method and machine learning model, it can be seen that RFE and SVM were appropriately fit.

The radiomic features used in this study were texture-based. It measures the roughness or smoothness of an image by defining the histogram of the image, measuring the intensity of the gray level, and defining its relationship with neighboring pixels. The maximum value was relatively low because there was no fracture in the normal state and no empty space between the bones. However, when there is a fracture, the value is relatively high, owing to the empty space and widening of the sacral fracture. The GLNN indicates the similarity of the intensity values at the gray level. This similarity is normal because there is no curvature or roughness owing to the fracture, and the brightness of the image is uniform. In the fracture type, the closer to Type C, the wider the ceiling joint, and rotation and vertical instability of the pelvic ring exist; therefore, the brightness value of the image is not uniform. Minimum, like Maximum, was relatively low in normal fractures and relatively high in fracture types. GLN, like GLNN, measures the similarity at the gray level. MaximumProbability refers to the largest pixel pair among adjacent pixels. Normally, the number of fractures is relatively low compared to the fracture types. In the fracture type, it is relatively high as an empty space because of fracture or widening. Fractures have a higher image texture complexity than normal fractures.

The radiomics features obtained from the research results are quantitative, unlike the visual analysis methods of conventional doctors. Visual analysis by conventional doctors has clear criteria but no quantitative values. These quantitative values are obtained based on the texture of the X-ray image and are difficult to check with the human eye. The texture of an X-ray image defines the relationship between neighboring pixels. Therefore, we obtained radiomics feature values to classify the fracture type by utilizing the quantitative values. We also found that the reproducibility of radiomics features is relatively high. In the future, training and testing on multi-institutional data will increase the reliability of the acquired features.

## Conclusions

Previous studies have reported the diagnosis of fractures as well as osteoporosis using these features [30–32]. These studies have attempted to detect the presence or absence of fractures or osteoporosis using each feature; however, the present study attempts to classify pelvic ring fractures by distinguishing different features of the fracture surface or line according to the morphological characteristics of each bone constituting the pelvic ring. To the best of our knowledge, such studies have not been conducted previously. Although there have been studies on the diagnosis of fractures, these are of great significance because studies classifying fracture patterns have not been conducted.

However, the experimental results presented herein have certain limitations. First, this study has limitations as a retrospective study. Second, because the analysis was performed

using pelvic X-ray images, there were cases where the fracture site was not clearly indicated because of bowel gas located in the pelvis or other organs in the pelvis. Therefore, the posterior surface of the pelvic ring, that is, the sacral fracture or sacroiliac joint widening, may not have been clearly analyzed. Third, AO/OTA classification classifies fracture patterns based on the location of the fracture and the vertical and rotational instability of the pelvic ring; however, radiomics-based machine learning analysis has limitations in reflecting three-dimensional information, such as vertical and rotational instability. Fourth, the elderly patients may have developed osteoporosis. In such cases, the texture of the fracture in the X-ray image may differ. In this study, because the analysis was performed regardless of age, care must be taken when interpreting or generalizing the results.

Unlike with existing classification methods, this study was able to classify normal pelvis and pelvic fractures of types A, B, and C, as suggested by AO/OTA classification, through feature selection and machine learning algorithms. We identified ten features that contributed to this classification. It is thought that a study to classify and extract the features of each bone by dividing each bone constituting the pelvic ring in more detail is needed. If the performance is improved through additional research, it will be of great help in developing a pelvic fracture diagnostic aid system.

## Author Contributions

**Data curation:** Kwang Gi Kim, Gil Jae Lee.

**Formal analysis:** Seung Hwan Lee.

**Methodology:** Young Jae Kim, Kwang Gi Kim.

**Project administration:** Young Jae Kim.

**Resources:** Seung Hwan Lee, Gil Jae Lee.

**Validation:** Jun Young Park.

**Writing – original draft:** Jun Young Park.

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
