## [Decision Letter · Decision Letter 0]

8 Mar 2024

PONE-D-23-42660Machine Learning Model Based on Radiomics Features for AO/OTA Classification of Pelvic Fractures on Pelvic RadiographsPLOS ONE

Dear Dr. Kim,

Thank you for submitting your manuscript to PLOS ONE. After careful consideration, we feel that it has merit but does not fully meet PLOS ONE’s publication criteria as it currently stands. Therefore, we invite you to submit a revised version of the manuscript that addresses the points raised during the review process.

We look forward to receiving your revised manuscript.

Kind regards,

Barry Kweh

Academic Editor

PLOS ONE

Journal Requirements:

3. Thank you for stating the following financial disclosure: "This work was supported by the Korea Medical Device Development Fund grant funded by the Korea government (the Ministry of Science and ICT, the Ministry of Trade, Industry and Energy, the Ministry of Health & Welfare, the Ministry of Food and Drug Safety) (Project Number: 1711196789, RS-2023-00252804)."

4. Thank you for stating the following in the Acknowledgments Section of your manuscript: "This work was supported by the Korea Medical Device Development Fund grant funded by the Korea government (the Ministry of Science and ICT, the Ministry of Trade, Industry and Energy, the Ministry of Health & Welfare, the Ministry of Food and Drug Safety) (Project Number: 1711196789, RS-2023-00252804)."

Please remove any funding-related text from the manuscript and let us know how you would like to update your Funding Statement. Currently, your Funding Statement reads as follows: "This work was supported by the Korea Medical Device Development Fund grant funded by the Korea government (the Ministry of Science and ICT, the Ministry of Trade, Industry and Energy, the Ministry of Health & Welfare, the Ministry of Food and Drug Safety) (Project Number: 1711196789, RS-2023-00252804)."

5. In the online submission form, you indicated that the data used to support the findings of this study are available upon request from the corresponding authors.

Additional Editor Comments:

A study which attempts to use machine-learning in the radionomic classification of pelvic fractures. The authors have utilized X-Rays to classify fractures rather than computed topography which should be justified. A broader review of the literature in a tabulated and written format of using fracture line and morphological features would also be useful to the audience.

Reviewers' comments:

Reviewer's Responses to Questions

**Comments to the Author**

1. Is the manuscript technically sound, and do the data support the conclusions?

Reviewer #1: Partly

Reviewer #2: Yes

2. Has the statistical analysis been performed appropriately and rigorously? 

Reviewer #1: Yes

Reviewer #2: Yes

3. Have the authors made all data underlying the findings in their manuscript fully available?

Reviewer #1: No

Reviewer #2: Yes

4. Is the manuscript presented in an intelligible fashion and written in standard English?

Reviewer #1: Yes

Reviewer #2: Yes

5. Review Comments to the Author

Reviewer #1: The paper titled "Machine Learning Model Based on Radiomics Features for AO/OTA Classification of Pelvic Fractures on Pelvic Radiographs" presents a study aimed at developing a machine-learning algorithm based on radiomics for quick diagnosis and classification of pelvic fractures in X-ray images. The study involved analyzing pelvic anteroposterior radiographs from 990 adults with pelvic fractures and 200 normal subjects. A total of 93 features were extracted and analyzed using various machine learning models and feature selection methods.

I have a few concerns

The authors state the aim is to develop radiomics-based machine-learning algorithm to rapidly diagnose fractures in pelvic X- ray images and classify pelvic instability.

The results of the study do not support this aim.

2. How was the ROI defined, the extent and the software used?

3. There are 3 classes described and ROC is provided as the metric for model performance. Are the results supporting One vs Rest or One vs One class?

4. How was the radiomics models better than the clinician's decision?

5. Please do a thorough check before submission that only tables and figures included are cited appropriately.

Reviewer #2: This article describes the performance of machine learning models using radiomic features in classifying pelvic fractures on radiographs. The model using RFE method of feature selection and SVM classifier performed with the highest AUC of 0.75 +/- 0.06. Most models using RFE feature selection performed with AUC between 0.73 and 0.74, other methods of feature selection performed with variable AUC between 0.59 and 0.73. The importance values of features were also described.

It is the first study to evaluate the performance of machine learning models in classification of pelvic fractures and describe the significant radiomics features. Other studies have used machine learning models to classify hip fractures with high accuracy.

6. PLOS authors have the option to publish the peer review history of their article (what does this mean?). If published, this will include your full peer review and any attached files.

Reviewer #1: No

Reviewer #2: No

---

## [Author Response · Author response to Decision Letter 0]

22 Apr 2024

Editor

Answer

As the editor mentioned, it appears that the format of this paper did not fit the requirements defined by PLOS ONE. Therefore, the authors revised and supplemented the overall format of the paper by referring to the PDF files mentioned by the editor in the email. Thanks to the editor's comments, the completeness of the paper has improved. Thank you.

Answer

As the editor mentioned, we believe there was a lack of submissions supporting the results of this study. Evidence that supports research results is essential and can improve the reliability of experimental results. Therefore, the authors uploaded the entire code and virtual environment used in the experiment to GitHub. Please refer to the path below.

https://github.com/user-dynamite/biomedical-engineering/tree/master

3. Thank you for stating the following financial disclosure: "This work was supported by the Korea Medical Device Development Fund grant funded by the Korea government (the Ministry of Science and ICT, the Ministry of Trade, Industry and Energy, the Ministry of Health & Welfare, the Ministry of Food and Drug Safety) (Project Number: 1711196789, RS-2023-00252804)."

Answer

Thank you for providing funding and for carefully reviewing and confirming the details of the paper. The funders who provided funding for this paper had no role in the preparation of the manuscript. A description of the role of the funder was included in the cover letter. Please confirm.

 

4. Thank you for stating the following in the Acknowledgments Section of your manuscript: "This work was supported by the Korea Medical Device Development Fund grant funded by the Korea government (the Ministry of Science and ICT, the Ministry of Trade, Industry and Energy, the Ministry of Health & Welfare, the Ministry of Food and Drug Safety) (Project Number: 1711196789, RS-2023-00252804)."

Please remove any funding-related text from the manuscript and let us know how you would like to update your Funding Statement. Currently, your Funding Statement reads as follows: "This work was supported by the Korea Medical Device Development Fund grant funded by the Korea government (the Ministry of Science and ICT, the Ministry of Trade, Industry and Energy, the Ministry of Health & Welfare, the Ministry of Food and Drug Safety) (Project Number: 1711196789, RS-2023-00252804)."

Answer

We are fully aware of the failure to carefully check the details of the funds statement in the PLOS ONE form. Thanks to the Editor's comments, we were able to check and review the matter in detail. All text related to funds was deleted from the manuscript. The contents of the fund statement were revised and supplemented and written in the cover letter.

5. In the online submission form, you indicated that the data used to support the findings of this study are available upon request from the corresponding authors.

Answer

Sharing data used in research, such as the editor's comments, increases the possibility of further research and future development. This research team is fully aware of the content and believes it is important. However, it is impossible for Gachon University Gil Hospital, which received the patient data, to disclose the patient data in a public repository. If you contact the corresponding author individually, the corresponding author can share data personally with institutional approval. We ask for the Editor’s generous understanding regarding this content.

 

Reviewer 1

The paper titled "Machine Learning Model Based on Radiomics Features for AO/OTA Classification of Pelvic Fractures on Pelvic Radiographs" presents a study aimed at developing a machine-learning algorithm based on radiomics for quick diagnosis and classification of pelvic fractures in X-ray images. The study involved analyzing pelvic anteroposterior radiographs from 990 adults with pelvic fractures and 200 normal subjects. A total of 93 features were extracted and analyzed using various machine learning models and feature selection methods.

1. The authors state the aim is to develop radiomics-based machine-learning algorithm to rapidly diagnose fractures in pelvic X- ray images and classify pelvic instability.

The results of the study do not support this aim.

Answer

As the reviewer commented, this paper developed a radiomics-based machine learning learning model and achieved a performance of about AUC 0.75. In this paper, pelvic instability was expressed in quantitative numbers based on the features selected as important. Recent studies have achieved AO/OTA pelvic fracture classification performance of up to Sensitivity 75.7%, Precision 83.6%, and Accuracy 85.0% using ResNeXt50, RNN, and 3D-ResNet50 in CT images [1]. By using Inception-V3 on in this way, the type of fracture was classified by learning a deep learning model, but it was not expressed in quantitative numbers. They did not perform any better than medical specialists. The field is developing artificial intelligence models in various aspects to assist medical professionals in their diagnosis. The authors also believe that if additional research is conducted based on the selected characteristics, it will be able to assist medical professionals in their diagnosis. Thank you for mentioning important points to improve the completeness of this paper.

References

[1] Dreizin, David, et al. "An automated deep learning method for tile AO/OTA pelvic fracture severity grading from trauma whole-body CT." Journal of Digital Imaging 34 (2021): 53-65.

[2] Lee, Changhwan, et al. "Classification of femur fracture in pelvic X-ray images using meta-learned deep neural network." Scientific reports 10.1 (2020): 13694.

2. How was the ROI defined, the extent and the software used?

Answer

As the reviewer commented, I felt that this paper lacked mention of this topic. In this study, the ROI corresponding to the pelvic region was defined by referring to the findings of pelvic AP X-rays and CT images by a trauma surgeon with more than 10 years of experience. Additionally, the defined ROI includes the left and right ilium, pubic bone, and ischium. The software used to obtain ROI was the commercial software AVIEW (Corelinesoft, Seoul, Republic of Korea). This content has also been added to the paper. Thank you for leaving a comment to improve the completeness of this paper.

Modified and supplemented contents - Page 5, line 3~7

 

3. There are 3 classes described and ROC is provided as the metric for model performance. Are the results supporting One vs Rest or One vs One class?

Answer

This paper classified the models into four classes (Type A, Type B, Type C, Normal) and compared the performance between models by obtaining AUC values from the ROC curve. The results of this study support One vs Rest. One vs Rest turns a multi-class problem into several binary classification problems, and a binary classifier is learned by treating one class as 1 and the remaining classes as 0. This method can achieve higher performance than One vs One. In the case of One vs One, the data is split into two different classes and trained with a binary classifier. This means that as the number of classes increases, the classifier increases and becomes more sensitive to imbalances between classes. We chose One vs Rest to achieve higher performance with fewer classifiers. Added information about using One vs Rest. Thank you for improving the completeness of this paper by leaving a comment regarding the shortcomings of this paper.

Modified and supplemented contents - Page 7, line 24~26

4. How was the radiomics models better than the clinician's decision?

Answer

This paper conducted a study to classify pelvic instability using a model learned based on radiomics. The results provided in this paper do not compare or analyze the decisions of medical experts and the model's predicted results. Conventional medical specialists had standards for classifying pelvic instability, but there were no quantitative figures for instability classes. In the results of this paper, 10 radiomics features were selected as relatively important features, as shown in Figure 8. Through the results of the study, we were able to obtain quantitative figures for the class. We also found that the research results were highly reproducible. If we study multi-center data and conduct additional research in the future, we believe it will be at a level that can assist medical specialists in their diagnosis.

5. Please do a thorough check before submission that only tables and figures included are cited appropriately.

Answer

All tables and figures included in this paper are cited to support the content of the paper. Figure 1 is a pelvic AP X-ray image and an ROI mask image designating the pelvic region. Figure 2 shows an actual AP X-ray image of the pelvis where the pelvic area is not clearly visible due to organs or gas. These cases were excluded because they could be a hindrance when extracting radiomics features. Figure 3 is the result of performing the Histogram Equalization algorithm to make somewhat blurry pelvic AP X-ray images clear. The horizontal axis of the graph below in Figure 3 represents the contrast value of the image, and the vertical axis represents the frequency with which that contrast value is used. Figure 4 shows a flow chart for extracting radiomics features of the pelvic region from pelvic AP X-ray images. A total of 93 radiomics features were extracted. Figure 5 includes the statistical analysis of the machine learning model prediction results from the feature selection method algorithm after radiomics feature extraction. Figure 6 is a Heatmap graph obtained by obtaining the AUC of each combination using 7 machine learning models and 4 feature selection method algorithms. You can check the AUC of the combination in color and number at once. Figure 7 is the ROC curve for each combination of RFE feature selection method and each machine learning model. The ROC curve was obtained by selecting only the RFE feature selection method because RFE showed relatively high performance compared to other feature selection methods. Figure 8 shows the average feature importance value of each machine learning model combination for the 10 features extracted using the RFE feature selection method. Figure 9 shows the feature importance values of each machine learning model combination for the 10 features extracted using the RFE feature selection method. Table 1 shows pelvic AP X-ray images obtained from 2015 to 2020 by type. Table 2 is the distribution of Train set and Test set for each type. There are a total of 11 figures and tables, which may seem a bit large, but they are all written to support the content of the paper. Thanks to the reviewer's comments, we were able to thoroughly check and review the figures and tables cited in the paper. The completeness of the paper has been improved thanks to the reviewer's comments. Thank you.

Reviewer 2

This article describes the performance of machine learning models using radiomic features in classifying pelvic fractures on radiographs. The model using RFE method of feature selection and SVM classifier performed with the highest AUC of 0.75 +/- 0.06. Most models using RFE feature selection performed with AUC between 0.73 and 0.74, other methods of feature selection performed with variable AUC between 0.59 and 0.73. The importance values of features were also described.

It is the first study to evaluate the performance of machine learning models in classification of pelvic fractures and describe the significant radiomics features. Other studies have used machine learning models to classify hip fractures with high accuracy.

Answer

First, thank you very much for leaving a positive comment on this paper. In this paper, we developed a radiomics-based machine learning model to classify pelvic instability. Using the prediction results, performance was evaluated and important radiomics characteristics were analyzed and explained. As the reviewer commented, our researchers believe that this is the first study to classify pelvic instability based on radiomics features. I think it is meaningful as a first-of-its-kind study, and I think the extracted important radiomics features can be used as biomarkers in the future. Thank you.

---

## [Editor Report · Decision Letter 1]

25 Apr 2024

PONE-D-23-42660R1Machine learning model based on radiomics features for AO/OTA classification of pelvic fractures on pelvic radiographs.PLOS ONE

Dear Dr. Kim,

Thank you for submitting your manuscript to PLOS ONE. After careful consideration, we feel that it has merit but does not fully meet PLOS ONE’s publication criteria as it currently stands. Therefore, we invite you to submit a revised version of the manuscript that addresses the points raised during the review process.

We look forward to receiving your revised manuscript.

Kind regards,

Barry Kweh

Academic Editor

PLOS ONE

Journal Requirements:

**Additional Editor Comments:**

A well written article which has clarified its methodology in response to the reviewers. A further refinement is a broader discussion of the literature and a tabulated as well as written summary of existing radiomic features and key pelvic fracture findings should be provided to authors to further strengthen the authors' findings.

---

## [Author Response · Author response to Decision Letter 1]

7 May 2024

Journal Requirements

Answer

It is important to write references completely and accurately as required by the journal. We, the authors, fully agree with the journal's requirements, as the references are what support the research in the paper. We have checked and reviewed them thoroughly. We have reviewed the references of the papers and none of them have been retracted. Thank you for your comments to improve the quality of the paper.

Additional Editor Comments

1. A well written article which has clarified its methodology in response to the reviewers. A further refinement is a broader discussion of the literature and a tabulated as well as written summary of existing radiomic features and key pelvic fracture findings should be provided to authors to further strengthen the authors' findings.

Answer

First, thank you for your positive comments on this paper. Thanks to the editors and reviewers, the quality of the paper seems to have improved a lot. Additionally, as the editor comment, a more extensive discussion of the paper and existing radiomics features and findings of major pelvic fractures would have strengthened the results. We completely agree with the editor's opinion. We have included a full discussion of the paper and a summarized version of the broader results in the final paragraph of Results and Discussion. We believe that it is essential to define the presentation of preexisting pelvic fractures. Pelvic fracture classification was added to the data section of the Materials and Methods section, as shown in Table 1. Because this is the basis from which we got our data. Table 1 is based on reference [12]. However, in our review, previous studies have not found radiographic features to classify pelvic fracture type. Thank you for mentioning important points to improve the paper.

Modified and supplemented contents - Page 11, line 25~32 / Page 4, line 16~19

---

## [Editor Report · Decision Letter 2]

10 May 2024

Machine learning model based on radiomics features for AO/OTA classification of pelvic fractures on pelvic radiographs.

PONE-D-23-42660R2

Dear Dr. Kim,

We’re pleased to inform you that your manuscript has been judged scientifically suitable for publication and will be formally accepted for publication once it meets all outstanding technical requirements.

Kind regards,

Barry Kweh

Academic Editor

PLOS ONE

Additional Editor Comments (optional):

An interesting article based on the radiomics of pelvic fractures. The discussion has been expanded and tables addressed.
---

## [Editor Report · Acceptance letter]

17 May 2024

PONE-D-23-42660R2 

PLOS ONE

Dear Dr. Kim, 

I'm pleased to inform you that your manuscript has been deemed suitable for publication in PLOS ONE. Congratulations! Your manuscript is now being handed over to our production team.

Kind regards, 

on behalf of

Dr. Barry Kweh 

Academic Editor

PLOS ONE